# Amino Acid Sequences of Lactoferrin from Red Deer (*Cervus elaphus*) Milk and Antimicrobial Activity of Its Derived Peptides Lactoferricin and Lactoferrampin

**DOI:** 10.3390/foods10061305

**Published:** 2021-06-07

**Authors:** Ye Wang, James D. Morton, Alaa EL-Din A. Bekhit, Alan Carne, Susan L. Mason

**Affiliations:** 1Department of Wine, Food and Molecular Biosciences, Lincoln University, P.O. Box 84, Lincoln 7674, New Zealand; james.morton@lincoln.ac.nz (J.D.M.); sue.mason@lincoln.ac.nz (S.L.M.); 2Department of Food Science, University of Otago, P.O. Box 56, Dunedin 9054, New Zealand; aladin.bekhit@otago.ac.nz; 3Department of Biochemistry, University of Otago, P.O. Box 56, Dunedin 9054, New Zealand; alan.carne@otago.ac.nz

**Keywords:** deer lactoferrin amino acids, deer lactoferricin, deer lactoferrampin, antibacterial activity

## Abstract

Although the bioactivities of bovine lactoferrin have been extensively investigated, little is known about deer milk lactoferrin bioactivity and its amino acid sequence. This research investigated the amino acid sequence of deer lactoferrin and the antimicrobial activities of two lactoferrin-encrypted peptides; lactoferricin (Lfcin) and lactoferrampin (Lfampin). Deer lactoferrin was found to have a molecular weight of 77.1 kDa and an isoelectric point of 7.99, which are similar to that of bovine lactoferrin, 78 kDa and pI 7.9. Deer lactoferrin contains 707 amino acids, one amino acid less than bovine lactoferrin, and has 92% homology with bovine lactoferrin. Deer lactoferricin exhibited strong antimicrobial activity against *E. coli* American Type Culture Collection (ATCC) 25922 and *L. acidophilus* ATCC 4356. The antimicrobial activities of deer and bovine Lfcin and Lfampin were compared. Based on MIC, deer Lfcin was found to be a more effective inhibitor of *L. acidophilus* ATCC 4356 than bovine Lfcin, but bovine Lfcin and Lfampin were more effective against *E. coli* ATCC 25922 than deer Lfcin and Lfampin. The deer Lfcin sequence differed at seven amino acids from bovine Lfcin and this decreased the net positive charge and increased the hydrophobicity. Deer Lfampin contained two differences in amino acid sequence compared to bovine Lfampin which decreased the net positive charge. These amino acid sequence differences likely account for differences in antibacterial activity. Positive charge and hydrophobic residues provide the amphipathic character of these helical peptides, and are considered important for binding of antimicrobial peptides. In silico modelling of deer Lfcin indicated an identical α-helical structure compared to bovine Lfcin.

## 1. Introduction

The extended use of antibiotics has led to antibiotic resistance which threatens global health and food security. Antimicrobial peptides have potential as alternatives to antibiotics. Compared with traditional antibiotics, a typical antimicrobial peptide can have a higher isoelectric point, better thermal stability, broader-spectrum antimicrobial properties, and can be more amenable to preparation in a sterile form [1]. It has been suggested that antimicrobial peptides should not cause widespread resistance, due to their preferential attack on bacterial cell membranes [2]. Dairy products are an important source of antimicrobial peptides [3]. There has been a dramatic increase in the farming of red deer recently and emergence of deer milk dairying, hence deer milk could provide a potential alternative source of antimicrobial peptides [4,5].

The whey protein lactoferrin (Lf), an iron-binding glycoprotein of the transferrin family, exhibits antimicrobial properties in its intact form. Lf from bovine is the most studied among all species [6]. Bovine Lf is comprised of 708 amino acids with a molecular weight of 78 Da [7]. A number of mechanisms have been suggested for the antimicrobial activity of Lf, including iron deprivation from bacteria [8], direct interaction with pathogenic microorganisms causing cell lysis [9], and prevention of the interaction between bacteria and host cells [10]. Several antimicrobial peptides are reported to be encrypted in bovine Lf. The most well-known are the two peptides, lactoferricin (Lfcin, 17–41) and lactoferrampin (Lfampin, 265–284). These two peptides are reported to exhibit substantial broad spectrum activity against bacteria, yeast, fungi and parasites [11]. Bovine Lfcin derived from the N-terminal domain of Lf (17–41) (FKCRRWQWRMKKLGAPSITCVRRAF) [12] can be generated by in vitro digestion with pepsin at pH 3, 37 °C for 4 h, and was found to display broad spectrum activity against bacteria, yeast, fungi, and parasites [13]. Bovine Lfampin (268–284) (WKLLSKAQEKFGKMKSR), is derived from the C-terminal domain of Lf [12]. Bovine Lfampin has been chemically synthesized and demonstrated to exhibit candidacidal activity, and antibacterial activity against *Bacillus subtilis*, *Escherichia coli*, and *Pseudomonas aeruginosa* [14].

No studies to date have been carried out investigating the amino acid sequence of deer Lf. Analysis of the primary structure of antimicrobial peptides encrypted in proteins such as Lf can help in understanding the mechanisms of their antimicrobial activities. Comparison of the similarities and differences between deer Lf and bovine Lf may provide insights into the antimicrobial activity of deer Lf and its peptides, as well as any potential advantage of deer milk as an alternative milk product. The objectives of the present study were to investigate the amino acid sequence of Lfcin and Lfampin in deer milk Lf, compared to that of bovine Lf, and the antimicrobial properties of the peptides.

## 2. Materials and Methods

### 2.1. Materials

Ion exchange chromatography materials were obtained from GE Healthcare (now Cytiva), Uppsala, Sweden. All chemicals for sodium dodecyl sulphate–polyacrylamide gel electrophoresis (SDS-PAGE) were obtained from Bio-Rad Laboratories, Irvine, CA, USA, unless otherwise stated. Bacterial strains used in antibacterial activity assays (Escherichia coli ATCC 25922, Staphylococcus aureus ATCC 25923, and L. acidophilus ATCC 4356) were from ESR (Christchurch, Canterbury, New Zealand). A LIVE/DEAD BaclightTM bacterial viability kit (L7007) was from Molecular Probes, Eugene, OR, USA, with detection using a fluorescence microscope (Nikon Eclipse 50i, Tokyo, Japan), were used for assay of bacterial cell viability. Commercial bovine Lf was used for in vitro digestion and antibacterial activity assay.

### 2.2. Purification of Deer Lactoferrin by Fast Protein Liquid Chromatography

Lf was enriched from deer milk using a sequential ion exchange chromatography method [15]. Deer milk sweet whey was prepared by adding 1 mL renin solution (Renco, Eltham, New Zealand) per litre of deer milk, incubating at 37°C, 1 h, and the clarified sweet whey fraction was obtained by centrifuging at 10,000× *g*, 15 min. Deer milk whey was filtered (DVPP, 0.45 μm, Durapore^®^, Millipore, Carrigtwohill, Ireland) and the pH was adjusted to 3.8 with 1 M HCl and initially subjected to anion exchange chromatography using two × 5 mL HiTrap Q-FF cartridges (GE Healthcare) linked in series, equilibrated with 20 mM sodium citrate, pH 3.8, containing 40 mM NaCl, on a FPLC (BioLogic DuoFlow, GE Healthcare). The unbound protein fraction was collected and adjusted to pH 7.0 and loaded onto a 5 mL HiTrap SP-FF cartridge (GE Healthcare), equilibrated in 20 mM sodium dihydrogen phosphate, pH 7.0. Bound proteins were eluted with stepwise increase in ionic strength using sodium dihydrogen phosphate, pH 7.0, containing (i) 0.1 M NaCl, (ii) 0.4 M NaCl, and (iii) 1.0 M NaCl. Protein containing fractions were collected, concentrated, and desalted using Vivaspin^®^20, 10 kDa MWCO spin filter units (GE Healthcare). Fractions were analyzed by SDS-PAGE, to confirm that the Lf eluted in the 1.0 M NaCl step.

### 2.3. SDS-PAGE

Deer Lf was analyzed by SDS-PAGE on an in-house made 12% acrylamide gel run in a Bio-Rad mini-gel electrophoresis system according to a standard protocol [16]. The gels were made with resolving gel 1.5 M Tris (pH 8.8), stacking gel 0.5 M Tris (pH 6.8), 30% (*w*/*v*) acrylamide/bis-acrylamide (37.5:1, 2.6% C), 10% (*w*/*v*) SDS solution, using 10% (*w*/*v*) ammonium persulfate and TEMED as catalysts. Protein samples were diluted with sample buffer, which contained 0.23 M Tris (pH 6.8), 8% SDS, 40% (*v*/*v*) glycerol, 0.08% (*w*/*v*) bromophenol blue and 4% (*v*/*v*) mercaptoethanol, with heating at 72℃ for 10 min prior to loading on the gel. BioRad Precision Plus Protein^TM^ standard (#1610373) was used as a molecular weight marker. Electrophoresis was performed at a constant voltage of 120 V for 90 min. After electrophoresis, the gel was fixed in 50% (*v*/*v*) methanol and 7% (*v*/*v*) acetic acid for 15 min and then washed 3 × 5 min with reverse osmosis grade water and then stained with GelCode^TM^ Blue (Thermo Scientific^TM^, #24590, USA), according to the supplier’s instructions. Stained SDS gels were scanned using a CS90000F Mark II scanner (Canon, Auckland, New Zealand).

### 2.4. Deer Lf Amino Acid Sequence Analysis

The deer Lf band excised from a stained SDS-PAGE was subjected to an in-gel digestion process, in which the gel was diced into small pieces and subjected to in-gel digestion with sequencing grade modified trypsin (Promega, Madison, WI, USA) based on a published method [17]. The extracted tryptic peptides were dried in a Savant Speed Vac Plus (Savant, Hyannis, MA, USA), and then redissolved in 20–50 µL of 5% (*v*/*v*) acetonitrile, 0.2% (*v*/*v*) formic acid in Type 1 MQ-water, and aliquots were injected onto an Ultimate 3000 nano-flow uHPLC-System (Dionex, Sunnyville, CA, USA) that was in-line coupled to the nano-spray source of an LTQ-Orbitrap XL hybrid mass spectrometer (Thermo Scientific, San Jose, CA, USA). Peptides were separated on an in-house packed emitter-tip column (75 μm ID PicoTip fused silica tubing of length 8–9 cm (New Objective, Woburn, MA, USA), packed with C-18 resin, and using 0.2% (*v*/*v*) formic acid in MQ-water (buffer A) and 0.2% (*v*/*v*) formic acid in acetonitrile (buffer B), with a gradient developed from 5% (*v*/*v*) acetonitrile, 0.2% [*v*/*v*] formic acid to 99% [*v*/*v*] acetonitrile, 0.2% [*v*/*v*] formic acid in water at a flow rate of 400 µL/min [18].

A typical instrument setting for the LTQ-Orbitrap was full MS in a mass range between *m*/*z* 400–2000, performed in the Orbitrap mass analyzer with a resolution of 60,000 and an AGC target of 5e5. Preview mode for FTMS master scan was enabled to generate precursor mass lists. The highest 7 signals were selected for CID (collision induced dissociation)-MS/MS in the LTQ ion trap at a normalized collision energy of 35% using an AGC target of 1e6 and one micro scan. Dynamic exclusion was enabled with 2 repeat counts during 90 sec and an exclusion period of 120 sec. Exclusion mass width was set to 0.01. After the initial LC-MS analysis of the sample from SDS-PAGE, the LC-MS/MS was repeated twice with different LC gradients (45 min method = 5–25% B over 17 min, 25–40% B over 4 min, 40–99% B over 4 min) and (60 min method = 5–25% B over 27 min, 25–40% B over 8 min, 40–99% B over 4 min), to optimize peptide fractionation and maximize the dynamic range of peptide MS analysis by allowing for bias in hydrophobicity and complexity of the peptides in multiple fractions.

MS/MS data were searched against a deer protein sequence database that was generated from a deer transcriptome analysis undertaken in-house in the Department of Biochemistry, University of Otago. The Lf sequences were interrogated with the MS/MS data obtained using the Mascot search engine [19]. The search for the data obtained from the in-gel digest of the deer Lf protein band was set up for full tryptic peptides and a maximum 3 missed cleavage sites. The search for the data obtained from peptide fractions from RP-HPLC was set up for non-tryptic peptides. Deamidation, oxidized methionine, and carboxyamidomethyl cysteine, were included as variable modifications for in-gel digest samples, and deamidation and oxidized methionine were included for in-solution HPLC samples, with a fragment mass error of 0.4 Da and a precursor mass tolerance of 75 ppm. SwissProt was used to search for sequence coverage of deer Lf for tryptic, chymotryptic and semi-chymotryptic peptides.

### 2.5. Three-Dimensional Structures of Deer Lfcin and Lfampin In Silico Analysis

3-D structures of the peptides Lfcin and Lfampin were simulated in PyMOL (Schrödinger Inc., New York, NY, USA). Based on the amino acid sequences of deer Lf, 3-D structures of deer Lfcin and Lfampin were automatically modelled in minimized (optimized) energy form in PyMOL. The full amino acid sequences and the 3-D structure of bovine Lf were obtained from the Protein Data Bank (PDB ID: 1BLF) [12].

### 2.6. Antibacterial Activity Assay for Deer Lfcin and Lfampin

Synthetic deer and bovine Lfcin and Lfampin were custom made by GL Biochem Ltd. (Shanghai, China), according to their amino acid sequences. The commercial synthetic peptides were desalted and of high purity (>98%) and verified using HPLC and MS. The antibacterial activity assay for the synthetic peptides and controls was performed according to the European Committee for Antimicrobial Susceptibility Testing Protocol [20]. Bacteria in broth were serially diluted with broth to 5 × 10^5^ cells/mL for *E. coli* and *S. aureus*, and at 1.5 × 10^8^ cells/mL for *L. acidophilus*. This was because *L. acidophilus* at a lower cell density (10^5^ cells/mL) could not achieve experimentally a good growth curve. A 100 μL aliquot of bacteria in broth was transferred into each well of a 96-well plate and mixed with 100 μL peptide stock at different concentrations (15–960 µg/mL) prepared in broth. The OD600 of the cells in the 96-well plate was obtained hourly using a plate reader during 24/48 h incubation at 37 °C. Minimum inhibitory concentration (MIC) was defined as the lowest concentration of sample that caused complete inhibition of bacterial growth. It was determined based on a change in OD of ≤ 0.05 at 600 nm during 24/48 h incubation; for the determination of minimum bactericidal concentration (MBC), 100 μL sample was taken from wells where no growth was detected, and spread on agar plates and incubated at 37 °C for 48 h for bacterial viable count. The MBC was defined as the lowest concentration found where there was ≥ 99.9% loss of viable cells [21].

A positive antibiotic control was prepared with bacteria in 100 μL broth + 100 μL penicillin (10,000 units/mL) and streptomycin (10,000 µg/mL) (GIBCO 15140, Invitrogen^TM^, Waltham, MA, USA). A negative broth control was prepared with bacteria in 200 μL broth. A further control was uninoculated broth (200 μL) to test for sterility, all assays were performed in triplicate.

### 2.7. Bacterial Cell Viability by LIVE/DEAD BacLight^TM^ Bacterial Viability Kit

To distinguish between viable and non-viable bacterial cells following incubation with synthetic peptides, *E. coli* and *L. acidophilus* were stained using a LIVE/DEAD BacLight^TM^ Bacterial viability kit, according to the supplier’s instructions. Briefly, after 24 and 48 h incubation in an antibacterial activity assay, 200 µL of solution containing bacterial cells from wells containing synthesized peptides at minimum inhibitory concentration (MIC) were separately transferred into a 1.5 mL microfuge tube, washed three times in sterile 0.85% (*w*/*v*) NaCl and thoroughly mixed with Component B (SYTO^®^ 9 1.67 mM/propidium Iodide 18.3 mM), and incubated at room temperature in the dark for 15 min. The viability of bacteria was then observed using a Nikon Epi 50i fluorescence microscope (FM), equipped with a UV lamp and a 100× magnification objective. The epifluorescence microscopy counts were performed directly on the film adhered to the slide. The results were expressed as the average number of cells per microscope field of view and calculated from counting 25 microscope fields of view. FM was set according to the manufacturer’s description of excitation and emission profiles for the SYTO^®^ 9 and PI stains. The SYTO^®^ 9 green-fluorescent nucleic acid stain labels all bacteria, and the PI red-fluorescent nucleic acid stain only targets bacteria with damaged membrane. Using an appropriate mixture of these two dyes, bacteria with intact cytoplasm membrane stain fluorescent green, whereas bacteria with a damaged membrane stain fluorescent red.

### 2.8. Statistical Analysis

Minitab version 17 statistical software (Minitab Inc., State College, PA, USA) was used for data analysis. For bacterial cell viability, the mean ± SD were calculated from 25 fields from each slide. The data were subjected to one way analysis of variance (ANOVA), followed by the Sidak correction in General Linear Model to determine the significant differences between samples and broth control group, and intergroup comparisons, at *p* < 0.05 level.

## 3. Results

### 3.1. Chromatography Fractionation and Electrophoresis of Lf from Deer Milk

The chromatogram obtained from the BioLogic DuoFlow cation exchange chromatography, which was used to isolate Lf from deer milk, is shown in Figure 1. Deer Lf was eluted with 20 mM sodium dihydrogen phosphate buffer, pH 7.0, containing 1.0 M NaCl, corresponding to the third peak in Figure 1, and indicated that at pH 7, deer Lf was substantially positively charged, in requiring elevated ionic strength to be eluted from the cation exchange resin. In Figure 2, the molecular weight of Lf from deer milk was found to be in the vicinity of 75–80 kDa by SDS-PAGE. With an overloaded loading of the LF protein on the SDS-PAGE, an additional protein band is evident between 20–25 kDa. Analysis of this lower band by in-gel digest mass spectrometry did not generate an identity.

The unbound deer whey protein fraction (20 mL) obtained from the first chromatography FPLC step was adjusted to pH 7.0 and applied to a HiTrap SP-FF 5 × 5 mL cartridge equilibrated in 20 mM sodium dihydrogen phosphate at pH 7.0, at a flow rate of 5 mL/min. The Lf protein was eluted with 20 mM sodium dihydrogen phosphate, pH 7.0, containing 1 M NaCl.

### 3.2. Amino Acid Sequence of Deer Lactoferrin

The identification of the deer Lf obtained by chromatography was confirmed by in-gel digest mass spectrometry. The deer Lf amino acid sequence was obtained from a deer protein sequence database that was generated from a deer transcriptome analysis undertaken in-house in the Department of Biochemistry, University of Otago. The combined sequence coverage of tryptic, chymotryptic, and semi-chymotryptic peptides obtained was used to interrogate the in-house generated deer database and was found to achieve 65.21% coverage of the Lf amino acid sequence with matched peptides. The amino acid sequence of deer Lf which was generated from mRNA sequence is shown in Table 1, with comparison to the amino acid sequence of bovine Lf (PDB ID: 1BLF) [12]. Deer Lf was found to contain 707 amino acids, which is one amino acid less than the bovine Lf sequence. The primary structure of deer Lf was found to be 92% homologous with the bovine Lf sequence, with 55 differences in the amino acid sequence. The amino acid sequence of deer Lf reported in the present study, shows that it has a molecular weight of 77.1 kDa, that is lower than bovine Lf (78 kDa), and deer Lf has a slightly higher pI (7.99) than bovine Lf (7.9) [22].

#### Characteristics of Deer Lactoferricin and Lactoferrampin

Modelled 3-D structures of deer Lfcin and Lfampin were found to be similar to that of the bovine Lfcin and Lfampin structures (Figure 3). The amino acid sequences of Lfcin (17–41) from deer and bovine Lf share 72% homology. Deer Lfcin has seven positively charged amino acids and bovine Lfcin has eight positively charged amino acids. There are nine hydrophobic residues and nine hydrophilic residues in deer Lfcin. However, bovine Lfcin has two more hydrophobic residues than deer Lfcin. The high overall positive charge of peptides with a net charge of at least +4 and hydrophobic residues are reported to be important for antibacterial activity of bovine Lfcin [23,24]. Deer Lfcin possesses similar characteristics of positive charge and hydrophobic residues which indicate deer Lfcin should exhibit antibacterial activity like bovine Lfcin. But due to different numbers of positively charged amino acids and hydrophobic residues in deer Lfcin, it might have differences in binding to the bacterial cell wall, resulting in a difference in activity.

Deer and bovine Lfampin are homologous, apart from difference in two different amino acids, resulting in 90% similarity. Basic amino acids, Lys and Arg, in bovine Lfampin are replaced with amide amino acids in deer Lfampin. There are four positively charged amino acids in deer Lfampin and six in bovine Lfampin. Both peptides have seven hydrophobic residues.

### 3.3. Antibacterial Activities of Synthesized Peptides Lactoferricin and Lactoferrampin on Bacterial Growth

Deer and bovine Lfcin and Lfampin at different concentrations were tested against *E. coli* ATCC 25922, *S. aureus* ATCC 23923, and *L. acidophilus* ATCC 4356. The MIC of deer Lfcin against *E. coli* ATCC 25922 was found to be 240 µg/mL and the MBC was found to be the same (Table 2). Bovine Lfcin was found to be 120 µg/mL for both the MIC and the MBC for *E. coli* ATCC 25922 (Table 2). Deer Lfampin, within the concentration range from 15 to 480 µg/mL, promoted *E. coli* ATCC 25922 growth. The MIC, as well as MBC of bovine Lfampin against *E. coli* ATCC 25922 was found to be 480 µg/mL. These results indicate that at an equivalent concentration, bovine Lfcin and Lfampin were more active as antimicrobial inhibitors of *E. coli* ATCC 25922 growth than deer Lfcin and Lfampin.

Although no MICs were achieved within the tested concentrations (30–480 µg/mL) for deer or bovine Lfcin and Lfampin against *S. aureus* ATCC 25923, the lag time and generation time of *S. aureus* was found to increase in the presence of deer and bovine Lfcin at 240 and 480 µg/mL respectively. This indicates that both deer and bovine Lfcin slowed the growth of *S. aureus*. The MIC of deer Lfcin against *L. acidophilus* ATCC 4356 was found to be 480 µg/mL. At the same concentration, deer Lfcin showed more active antimicrobial inhibitory ability against *L. acidophilus* ATCC 4356 than did bovine Lfcin. Deer Lfampin was also more active against *L. acidophilus* ATCC 4356 than was bovine Lfampin at the same concentration, but no MICs were achieved.

The effects of deer Lfcin at 240 µg/mL, bovine Lfcin at 120 µg/mL, and bovine Lfampin at 480 µg/mL on *E. coli* ATCC 25922 viability at 0 and 24 h in the antimicrobial activity tests are shown in Figure 4. Green colored bacterial cells indicate an intact bacterial membrane, which at 0 h indicate that the bacteria were viable. After 24 h of exposure to antimicrobial peptide, the *E. coli* cells exhibited a red color. This indicated that *E. coli* ATCC 25922 bacteria were non-viable following exposure to the antimicrobial peptides. The antibiotics penicillin (10,000 units/mL) and streptomycin (10,000 µg/mL) were used as controls. Figure 4D shows the *E. coli* cell viability at 0 and 24 h in the presence of antibiotics. It can be concluded that these three peptides at their MICs had an equivalent inhibitory effect on *E. coli* ATCC 25922 as did the antibiotics.

*L. acidophilus* ATCC 4356 cell viability in the presence of deer Lfcin at 480 µg/mL at 0 and 24 h was compared with treatment with the antibiotics penicillin (10,000 units/mL) and streptomycin (10,000 µg/mL), along with an MRS broth control group. Figure 4F shows most of the *L. acidophilus* cells were viable (green) with only a few non-viable (red) at 0 h. After 24 h, although the number of bacterial cells increased, most of the bacteria were found to be non-viable following exposure to deer Lfcin at 480 µg/mL. With the antibiotic treatment, the bacterial viability at 0 h was the same as that of deer Lfcin in which most of the bacteria were viable, but at 24 h were non-viable, indicating that the antibiotics were more effective at inhibiting *L. acidophilus* growth than was deer Lfcin at 480 µg/mL. In Figure 4H for the MRSc broth control group, the majority of bacterial cells were viable at 0 h. After 24 h, there were more bacteria in the field and viable bacterial cells still outnumbered non-viable cells. Due to the variable dimensions of *L. acidophilus* that were observed in the microscope field, bacterial cell numbers could not be accurately determined.

## 4. Discussion

Lfcin, which is located in the N-domain of bovine Lf, can be produced by gastric pepsin cleavage of Lf [25]. It is a 25-amino-acid peptide that has a high proportion of basic residues, with a net charge of +8, and amphipathic properties [26]. The ability of Lfcin to form amphipathic structures with net hydrophobic and positively charged surfaces is a trait that is shared with other peptides having antimicrobial activity [27]. As Arg can interact both electrostatically and through multiple hydrogen bonds with the negatively charged surface of bacteria, it is thought that this amino acid is the most effective target in antimicrobial peptides to interact with the bacterial membrane [28]. Once this interaction occurs, the hydrophobic residues interact with the lipophilic portion of the membrane, becoming embedded into its surface and destabilizing the packing of the membrane phospholipids. Of the hydrophobic residues present in Lfcin, Trp is clearly important. The bactericidal activity of bovine Lfcin appears to be dependent on not just one Trp but on at least two [23]. Trp acts as an anchor with the bulk of Lfcin residues helping to bind the peptide to the membrane. The importance of Arg and Trp for the antibacterial activity of Lfcin highlights the importance of both electrostatic and hydrophobic interactions for the activity of Lfcin. The in silico modelling of deer Lfcin indicated a very similar α-helical structure compared to bovine Lfcin. Deer Lfcin has four Arg, one less than bovine Lfcin, that could interact with the bacterial membrane. One Arg in bovine Lfcin is replaced by Tyr in deer Lfcin. Both deer and bovine Lfcin have two Trp residues facilitating hydrophobic interaction with the bacterial membrane. The amino acid sequences predict that deer Lfcin could have similar antibacterial activity as bovine Lfcin.

Deer Lfampin has the common features of antimicrobial peptides with a high net positive charge, a hydrophobic domain and hence an amphipathic character. Although both Lfampin and Lfcin share amphipathic and cationic features, the bactericidal activity of Lfampin differs from that of Lfcin. Deer Lfampin has a different amino acid composition and chain length, hence its modelled structure differs somewhat from Lfcin. Bovine Lfampin is a more recently identified antimicrobial peptide compared to Lfcin and has been less studied in relation to its mechanism of antimicrobial activity. It has cationic amino acid residues with the hydrophobic domain containing Trp that is involved in membrane insertion [14]. The predicted amphipathic helical conformation of this peptide is a common motif for bioactivity against Gram-positive bacteria [29]. Trp 268 is reported to insert into the membrane at the lipid/water interface and the phenyl side chain of Phe 278 is oriented in the same direction as the indole ring of Trp 268, allowing these two residues to anchor the peptide into the lipid bilayer [29]. The amphipathic N-terminal helix is reported to anchor bovine Lfampin to the surface of the bacterial membrane with the hydrophobic side chains of Trp 268 and Phe 278 playing important roles in stabilizing the interaction and for inducing peptide folding [30]. Deer Lfampin 265–284 has both Trp 268 and Phe 278, which may be involved in membrane insertion during antimicrobial activity. Deer Lfampin Phe 278 could play an important role of interaction between the peptide and a negatively charged microbial cell membrane, while the helix capping residues, including Trp 268, could penetrate into the hydrophobic core of a membrane lipid bilayer.

Bovine Lfcin antimicrobial activity has been reported against *E. coli* ATCC 25922 within the range of MIC 3.3 to 30 µg/mL [11]. The variation of MIC of bovine Lfcin against *E. coli* could be related to the method used for determining the MIC, different broth, different strains, and source of Lfcin (pepsin hydrolyzed or chemically synthesized). Previously, simulated gastrointestinal hydrolysis of un-fractionated hydrolysates conducted in our laboratory, of deer Lf were found to exhibit differences in antibacterial activity compared to unfractionated hydrolysates of bovine Lf. In the present study, for synthetic deer Lfcin, the MIC against *E. coli* ATCC 25922 was found to be 240 µg/mL which was twice that of bovine Lfcin MIC in this study. Bovine Lfcin has more positive charged and hydrophobic residues than deer Lfcin. The difference in amino acid sequences of peptides suggests bovine Lfcin might have stronger antibacterial activity than deer Lfcin. This assumption was confirmed by the antimicrobial activity results, that indicated bovine Lfcin was more effective than deer Lfcin, at the same concentration, against *E. coli* ATCC 25922. Vorland et al. (1998) found the antimicrobial activity of bovine Lfcin (MIC 90 µg/mL) was more effective against *E. coli* ATCC 25922 than human (MIC > 200 µg/mL), murine (MIC >200 µg/mL), and caprine (MIC > 200 µg/mL) Lfcins [31]. These Lfcins were in linear form. Cyclic bovine Lfcin was found to be more active than linear bovine Lfcin against *E. coli* ATCC 25922 and *S. aureus* ATCC 25923. The effect of bovine Lfcin was decreased by reducing the cysteine disulfides. However, another report studying a fragment of bovine Lfcin consisting of 11 amino acids showed that the disulfide bridge is not essential for activity [32]. The motif RRWQWR is the minimum described sequence that exhibits antibacterial activity in bovine Lfcin. This motif is amphipathic, with the hydrophilic face being positive charged due to the presence of Arg side chains, while the hydrophobic face has Trp side chains [33]. Also, an even shorter motif RWQWR from bovine Lfcin was reported to exhibit a high and specific activity against *E. coli* ATCC 11775 [34]. The (R)RWQWR motif is encrypted in both deer Lfcin (SKCYRWQWRMKKLGTPFVTCVRRTS) and bovine Lfcin (FKCRRWQWRMKKLGAPSITCVRRAF), which could explain the antimicrobial activity exhibited by deer Lfcin found in the present study. Deer Lfcin contains positive charged and hydrophobic amino acid residues which are reported to be important for association and penetration of the bacterial membrane. More specifically, deer Lfcin contains the basic amino acid Arg which has the ability to interact with a negatively charged bacterial membrane, and two Trp that can enhance hydrophobic interaction of the peptide with a bacterial membrane.

The four peptides, Lfcin and Lfampin from deer and bovine, were not effective against *S. aureus* ATCC 25923 within the concentration range from 30 to 480 µg/mL in this study. It has been reported that the MIC of cyclic and linear bovine Lfcin against *S. aureus* ATCC 25923 was 30 and 90 µg/mL, respectively [31]. Different bacterial cell density, culture broth, and peptide structures may explain the different results obtained in the present study.

*L. acidophilus* ATCC 4356 was chosen as a representative of probiotic bacteria from human gut microbiota. Deer and bovine Lfcin and Lfampin were tested on *L. acidophilus* in order to investigate the effect of these antimicrobial peptides on bacteria which colonize the human gut. Deer Lfcin at 480 µg/mL reached the MIC against *L. acidophilus*. The present study showed the inhibitory activity of deer and bovine Lfcin and Lfampin on *L. acidophilus* growth. Chen et al. (2013) found *L. acidophilus* ATCC 4356 was inhibited by both bovine Lf and its hydrolysate with MIC 16 mg/mL for 24 h [35]. In the present study, deer and bovine Lfcin and Lfampin not only inhibited food-pathogenic bacteria (e.g., *E. coli* and *S. aureus*), but also probiotic bacteria.

The plate count method is routinely used in the determination of microbial viability. It is based upon the premise that a single bacterium can grow and divide to produce an entire colony. Microbial cells may exist in cryptobiotic, dormant, moribund, or latent states, in which they will not form colonies on nutrient media but may have other measurable activity [36]. No method exists that demonstrates with absolute certainty whether an organism is alive or dead. It is argued that the presence of intact and functional nucleic acids, as well as an intact and polarized cytoplasmic membrane are essential components of cellular viability. The membrane integrity stain LIVE/DEAD Baclight^TM^ is not a measure of life or death but assesses particular location-specific damage to cells. Both SYTO9 and PI stain nucleic acids. Green fluorescent SYTO9 generally labels all bacteria in a population, both those with intact membranes and those with damaged membranes. In contrast, red fluorescent PI which is a relatively large (668 Da) and double-charged dye, penetrates only bacteria with damaged cytoplasmic membranes, causing a reduction in the SYTO9 stain fluorescence when both dyes are present [37]. Thus, with an appropriate mixture of the SYTO9 and PI stains, bacteria with intact cell membranes stain fluorescent green, whereas bacteria with damaged membranes stain fluorescent red. The emission properties of the stain mixture bound to DNA change due to the displacement of SYTO9 by PI and quenching of SYTO9 emissions by fluorescence resonance energy transfer [38].

In addition to penetrating the cell wall and cell membrane of bacteria, bovine Lfcin can inhibit bacteria by affecting intracellular activity, by attacking the transcription-related activities and several cellular carbohydrate biosynthetic processes (DNA, RNA and protein synthesis), and can ultimately result in cell death in both Gram-negative and Gram-positive bacteria [39]. The present results of red fluorescence exhibited by *E. coli* ATCC 25922 and *L. acidophilus* ATCC 4356 after exposure to antimicrobial peptides for 24 h in the assays, showed that the bacterial cell membrane was damaged by deer Lfcin at 240 µg/mL, bovine Lfcin at 120 µg/mL, and bovine Lfampin and deer Lfcin at 480 µg/mL. Trp and Phe in bovine Lfampin have been reported to damage the integrity of the bacterial cell membrane [30], which was observed in the present study by the red fluorescence of *E. coli* ATCC 25922 treated with bovine Lfampin at 480 µg/mL.

## 5. Conclusions

Deer Lf was successfully fractionated from deer milk using cation exchange chromatography. Deer Lf was found to have a slightly higher isoelectric point (pI) of 7.99 compared to bovine Lf (pI 7.9). Deer Lf was found to contain 707 amino acids with a MW 77.1 kDa, with one less amino acid than bovine Lf (78 kDa). The deer Lf sequence was found to be 92% homologous to that of bovine Lf. Deer Lfcin and Lfampin share 72% and 90% similarities with bovine Lfcin and Lfampin, respectively. Deer Lfcin, bovine Lfcin and Lfampin exhibited strong inhibitory activity against *E. coli* ATCC 25922 with MICs of 240, 120, and 480 µg/mL respectively. Bovine Lfcin and Lfampin exhibited stronger antimicrobial activity than deer Lfcin and Lfampin against *E. coli* ATCC 25922 at the same concentrations. The MIC of deer Lfcin against *L. acidophilus* ATCC 4356 was 480 µg/mL in the present study. *L. acidophilus* ATCC 4356 was more susceptible to inhibition by deer Lfcin than bovine Lfcin. Although both deer Lfcin and Lfampin have amphipathic and cationic features, the antibacterial activities differ from that of bovine Lfcin and Lfampin, respectively, likely because they have different numbers of positive charged and hydrophobic residues which play important roles in the binding and penetration of the bacterial membrane in eliciting the antimicrobial activity.

## Figures and Tables

**Figure 1 foods-10-01305-f001:**
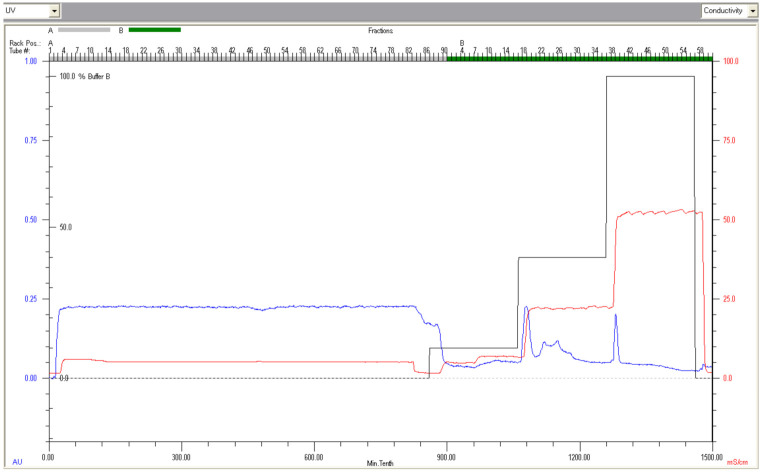
Fractionation of lactoferrin (Lf) from deer milk using cation exchange fast protein liquid chromatography (FPLC).

**Figure 2 foods-10-01305-f002:**
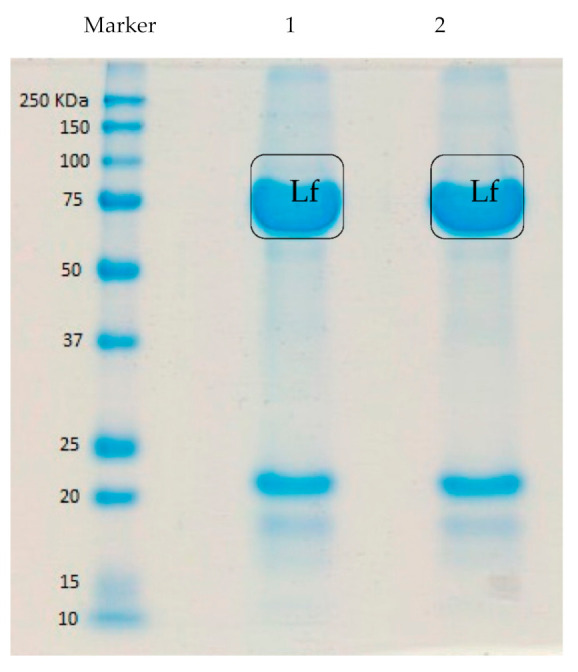
Sodium dodecyl sulfate-polyacrylamide acrylamide gel electrophoresis (SDS-PAGE) of lactoferrin (Lf) obtained by cation exchange chromatography from deer milk. Both lanes 1 and 2 are Lf isolated from deer milk.

**Figure 3 foods-10-01305-f003:**
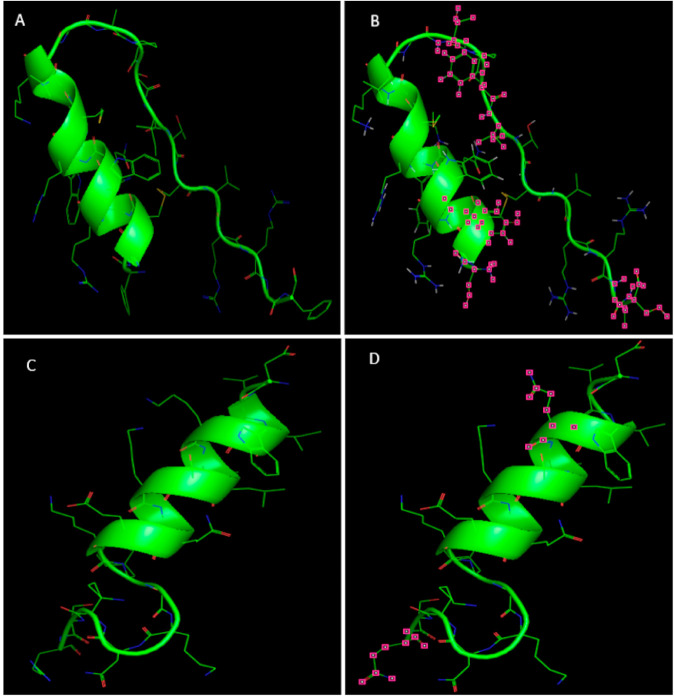
In silico structures of deer and bovine lactoferricin (Lfcin) and lactoferrampin (Lfampin). (**A**) Structure of bovine Lfcin (FKCRRWQWRMKKLGAPSITCVRRAF) (PDB ID: 1BLF); (**B**) Structure of deer Lfcin (SKCYRWQWRMKKLGTPFVTCVRRTS) displayed using PyMOL; (**C**) Structure of bovine Lfampin (265–284) (DLIWKLLSKAQEKFGKNKSR) (PDB ID: 1BLF); (**D**) Structure of deer Lfampin (265–284) (DLIWQLLSKAQEKFGKNKSQ) displayed using PyMOL. Pink squares indicate differences between deer and bovine Lfcin and Lfampin.

**Figure 4 foods-10-01305-f004:**
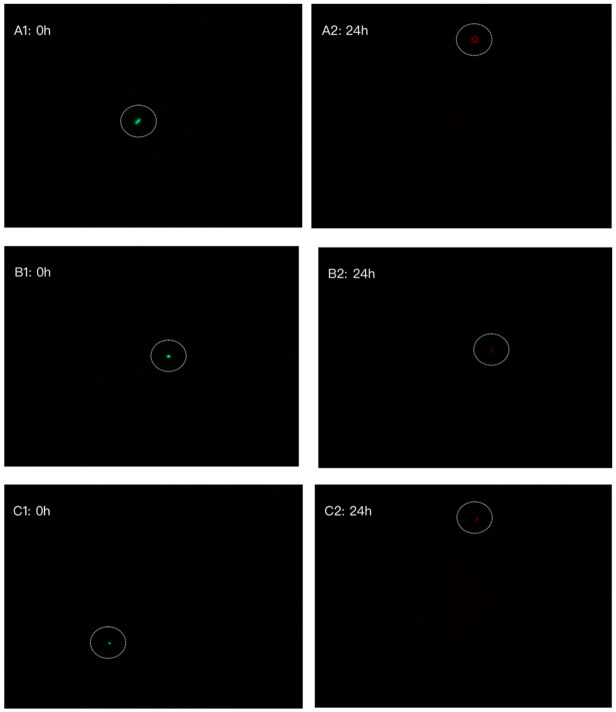
Bacterial viability determined using the *LIVE*/*DEAD* BacLight^TM^ Bacterial viability kit. Incubation with lactoferricin (Lfcin) and lactoferrampin (Lfampin) at 0 and 24 h. (**A**) *E. coli* ATCC 25922 incubated with deer Lfcin 240 µg/mL. (**B**) *E. coli* ATCC 25922 incubated with bovine Lfcin 120 µg/mL. (**C**) *E. coli* ATCC 25922 incubated with bovine Lfampin 480 µg/mL. (**D**) *E. coli* ATCC 25922 incubated with penicillin (10,000 units/mL) and streptomycin (10,000 µg/mL). (**E**) *E. coli* ATCC 25922 incubated with nutrient broth. (**F**) *L. acidophilus* ATCC4356 incubated with deer Lfcin 480 µg/mL. (**G**) *L. acidophilus* ATCC4356 incubated with penicillin (10,000 units/mL) and streptomycin (10,000 µg/mL). (**H**) *L. acidophilus* ATCC4356 incubated with MRSc broth. Green fluorescence indicates viable bacteria, red fluorescence indicates non-viable bacteria. Microscope magnification ×1000.

**Table 1 foods-10-01305-t001:** Comparison of the primary structures of deer lactoferrin (Lf) and bovine Lf.

Deer Bovine ^1^ 1–60	MKLFV PALL**A** LGALG LCLAA PRKNV RWC**A**I SQPEW **S**KC**Y**R WQWRM KKLG**T** P**FV**TC VRR**TS** ^3^ MKLFV PALLS LGALG LCLA ^2^ A PRKNV RWCTI SQPEW FKCRR WQWRM KKLGA PSITC VRRAF ^4^
Deer Bovine 61–120	ALECI RAIA**A** KKADA VTLD**S** G**L**VFE AG**L**DP YKLRP VAAEI YGT**EK** SPQTH YYAVA VVKKG ALECI RAIAE KKADA VTLDG GMVFE AGRDP YKLRP VAAEI YGTKE SPQTH YYAVA VVKKG
Deer Bovine 121–180	SNFQL DQLQG RKSCH TGLGR SAGW**N** IP**I**GI LRP**S**L **G**WTES LEPLQ GAVAK FFSAS CVPC**V** SNFQL DQLQG RKSCH TGLGR SAGWI IPMGI LRPYL SWTES LEPLQ GAVAK FFSAS CVPCI
Deer Bovine 181–240	D**GK**AY PNLCQ LCKG**T** GENQC ACS**P**R EPY_G YSGAF **R**CLQ**E** GAGDV AFVKE TTVFE NLPEK DRQAY PNLCQ LCKGE GENQC ACSSR EPYFG YSGAF KCLQD GAGDV AFVKE TTVFE NLPEK
Deer Bovine 241–300	ADRDQ YELLC LNNSR APVDA FKECH LAQVP SHAVV ARSVD GKEDL IW**Q**LL SKAQE KFGKN ADRDQ YELLC LNNSR APVDA FKECH LAQVP SHAVV ARSVD GKEDL IWKLL SKAQE KFGKN
Deer Bovine 301–360	KS**Q **^5^ SF QLFGS P**GS**QR DLLFK DSALG FLRIP SK**I**DS **E**LYLG **A**RYLT **A**LKNL RET**E**E EVKAR KSR ^6^ SF QLFGS PPGQR DLLFK DSALG FLRIP SKVDS ALYLG SRYLT TLKNL RETAE EVKAR
Deer Bovine 361–420	**S**TRVV WCAVG PEEQK KCQQW SQQS**D** Q**S**VTC ATAST TDDCI **A**LVLK GEADA L**S**LDG GYIYT YTRVV WCAVG PEEQK KCQQW SQQSG QNVTC ATAST TDDCI VLVLK GEADA LNLDG GYIYT
Deer Bovine 421–480	AGKCG LVPV**M** AENRK SSK**D**S SLDCV LRPTE GYLAV AVVKK ANEGL TWNSL K**G**KKS CHTAV AGKCG LVPVL AENRK SSKHS SLDCV LRPTE GYLAV AVVKK ANEGL TWNSL KDKKS CHTAV
Deer Bovine 481–540	DRTAG WNIPM GLI**A**N QTGSC **K**FDEF FSQSC APGAD PKS**S**L CALCA GDDQG LDKCV PN**T**KE DRTAG WNIPM GLIVN QTGSC AFDEF FSQSC APGAD PKSRL CALCA GDDQG LDKCV PNSKE
Deer Bovine 541–600	KYYGY TGAFR CLAED VGDVA FVKND TVWEN TNGES **S**ADWA KNLNR EDFRL LCLDG TRKPV KYYGY TGAFR CLAED VGDVA FVKND TVWEN TNGES TADWA KNLNR EDFRL LCLDG TRKPV
Deer Bovine 601–660	TEAQS CHLA**A** AP**S**HA VVSRS DRAAH V**E**QVL LHQQA LFG**R**N GK**D**CP DKFCL FKSET KNLLF TEAQS CHLAV APNHA VVSRS DRAAH VKQVL LHQQA LFGKN GKNCP DKFCL FKSET KNLLF
Deer Bovine 661–708	NDNT**R** CLAKL GGRPT YE**K**YL GTEYV TAIAN LKKCS TSPLL EACAF LTR NDNTE CLAKL GGRPT YEEYL GTEYV TAIAN LKKCS TSPLL EACAF LTR

Letters in bold indicate differences in the amino acid sequence in deer lactoferrin (Lf) compared to bovine Lf. ^1^ The amino acid sequence of bovine Lf was obtained from UniProtKB-P24627. ^2^ The first 19 amino acids of bovine Lf is signal peptide sequence that is cleaved at Ala19-Ala20 to yield the mature bovine Lf [7]. The numbering of the peptides below is with respect to the mature Lf sequence. ^3^ Underlined deer Lfcin f(17–41). ^4^ Underlined bovine Lfcin f(17–41). ^5^ Underlined deer Lfampin f(265–284). ^6^ Underlined bovine Lfampin f(265–284).

**Table 2 foods-10-01305-t002:** Minimum Inhibitory Concentrations (MIC) (µg/mL) of deer and bovine lactoferricin (Lfcin) and lactoferrampin (Lfampin) against *E. coli* ATCC 25922, *S. aureus* ATCC 25923, and *L. acidophilus* ATCC 4356. The Minimum Bactericidal Concentrations (MBC) values were found to be the same as that of the MIC values.

	Lfcin		Lfampin	
	Deer	Bovine	Deer	Bovine
*E. coli* ATCC 25922	240 ^b^	120 ^a^	>960 ^d^	480 ^c^
*S. aureus* ATCC 25923	>960	>960	>960	>960
*L. acidophilus* ATCC 4356	480 ^a^	>960 ^b^	>960 ^b^	>960 ^b^

^a–d^ means with different superscripts are significantly different at *p* < 0.05.

## Data Availability

The datasets generated for this study are available on request to the corresponding author.

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
