# Peer review of "Amino Acid Sequences of Lactoferrin from Red Deer (Cervus elaphus) Milk and Antimicrobial Activity of Its Derived Peptides Lactoferricin and Lactoferrampin"

_foods, 2021, doi:10.3390/foods10061305_

Round 1

Reviewer 1 Report

In the article entitled ,,Amino acid sequences of Lactoferrin from red deer (Cervus elaphus) milk and antibacterial activity of its derived peptides Lactoferricin and Lactoferrapin” the authors reported the bioactivities of individual peptides isolated from other source than bovine milk. Obtained results are very important, the results presented are relevant and the strategy is well established. I recommend the acceptance of the manuscript to be published in Foods, however after some minor revisions.

Lines 64-73 Please describe the isolation of lactoferrin from milk method with more details.

Lines 74-87 Please explain the usage of denaturation temperature equal to 72 °C instead of 95 °C; were the samples quantified before electrophoresis?

Lines 88-100 Please describe the digestion process with more details.

Line 138 – the bacteria density should be presented as 5 x 10^5 cell/mL; it should be corrected within the manuscript;

Line 141 – 100 μL instead of “One hundred μL”

Lina 207- mL

Figure 4 – I suggest to enlarge the photos, because they are almost out of the focus, but are worth to present to the reader; please, present scale bar on the photos.

Table 2 – Do the Authors possess comparable data for deer and bovine lactoferrin and could present it in the manuscript? These result could be compared in regard to the general antimicrobial potential; this information will greatly enrich the discussion of practical antibacterial activity of milk protein.

Author Response

Manuscript ID foods-1216765

Comments and Suggestions for Authors

1. In the article entitled “Amino acid sequences of Lactoferrin from red deer (Cervus elaphus) milk and antibacterial activity of its derived peptides Lactoferricin and Lactoferrapin” the authors reported the bioactivities of individual peptides isolated from other source than bovine milk. Obtained results are very important, the results presented are relevant and the strategy is well established. I recommend the acceptance of the manuscript to be published in Foods, however after some minor revisions.

We thank the Reviewer for their positive comment about our manuscript.

2. Lines 64-73 Please describe the isolation of lactoferrin from milk method with more details.

The following text has been added to the manuscript:

“Lf was enriched from deer milk using a sequential ion exchange chromatography method [12]. Deer milk sweet whey was prepared by adding 1 mL renin solution (Renco, Eltham, New Zealand) per litre of deer milk, and the clarified sweet whey fraction was obtained by centrifuging at 10,000 x g, 15 min. Deer milk whey was filtered (DVPP, 0.45 μm, Durapore®, Millipore, Carrigtwohill, Ireland) and the pH was adjusted to 3.8 with 1 M HCl and initially subjected to anion exchange chromatography using two × 5 mL HiTrap Q-FF cartridges linked in series, equilibrated with 20 mM sodium citrate, pH 3.8, containing 40 mM NaCl, on a FPLC (BioLogic DuoFlow, GE Healthcare, Auckland, New Zealand). The unbound protein fraction was collected and adjusted to pH 7.0 and loaded onto a 5 mL HiTrap SP-FF cartridge, equilibrated in 20 mM sodium dihydrogen phosphate, pH 7.0. Bound proteins were eluted with stepwise increase in ionic strength using sodium dihydrogen phosphate buffer, pH 7.0, containing (i) 0.1 M NaCl, (ii) 0.4 M NaCl and (iii) 1.0 M NaCl. Protein containing fractions were collected, concentrated and desalted using Vivaspin®20, 10 kDa MWCO spin filter units (GE Healthcare). Fractions were analysed by SDS-PAGE, to confirm that the Lf eluted in the 1.0 M NaCl eluted fraction.”

3. Lines 74-87 Please explain the usage of denaturation temperature equal to 72 °C instead of 95 °C; were the samples quantified before electrophoresis?

It is quite common when using commercial supplied materials for SDS-PAGE that the ‘manufacturers recommendation’ is to heat the protein samples with the SDS sample buffer at around 70oC (the Thermo Invitrogen BOLT SDS-PAGE system also has this recommendation). It has been found that 70oC is sufficient for denaturation of the protein and coating with SDS.

4. Lines 88-100 Please describe the digestion process with more details.

The following text has been added:

“Deer Lf band excised from stained SDS-PAGE was subjected to an in-gel digestion process in which the gel was diced into small pieces and subjected to in-gel digestion with sequencing grade modified trypsin (Promega, USA) based on a published method.”

5. Line 138 – the bacteria density should be presented as 5 x 10^5 cell/mL; it should be corrected within the manuscript;

Revised.

6. Line 141 – 100 μL instead of “One hundred μL”

Revised.

7. Line 207- mL

Revised.

8. Figure 4 – I suggest to enlarge the photos, because they are almost out of the focus, but are worth to present to the reader; please, present scale bar on the photos.

Done.

9. Table 2 – Do the Authors possess comparable data for deer and bovine lactoferrin and could present it in the manuscript? These result could be compared in regard to the general antimicrobial potential; this information will greatly enrich the discussion of practical antibacterial activity of milk protein.

Simulated gastrointestinal hydrolysis (to mimic digestion in the gut on consumption) previously performed in our lab found that un-fractionated hydrolysates of deer Lf exhibited differences in antibacterial activity compared to un-fractionated hydrolysates of bovine Lf.

A comment has been added to the Discussion section of the manuscript in relation to this.

Reviewer 2 Report

Referee report

Manuscript: foods 1216765

This manuscript entitled “Amino acid sequences of Lactoferrin from red deer (Cervus elaphus) milk and antibacterial activity of its derived peptides Lactoferricin and Lactoferrapin” aims to compare: i) The primary structure between deer and bovine lactoferrin (Lf); ii) The characteristics of modelled 3-D structures of Lactoferricin (Lfcin) and Lactoferrapin (Lfampin) peptides obtained from deer and bovine Lf and; iii) the antibacterial activities of Lfcin and Lfampin peptides obtained from deer and bovine Lf.

The originality and relevance of this study was well exposed and consisted in the fact that “No studies to date have been carried out investigating the amino acid sequence of deer Lf” and the antibacterial activity of it derived peptides Lfcin and Lfampin. The topic was well introduced (but can be improved), the procedure clearly described, and results were properly presented in tables and graphs and adequately discussed, using several updated bibliographic references.

The discussion and the conclusions were well described.

Based on these judgments, I recommend the manuscript for publication in Foods after minor revisions.

L19-21: In the Abstract section, the conclusion can be stronger if the authors present the reason for concluding that the differences of antibacterial activity against E. coli 20 ATCC 25922 of the peptides (Lfcin and Lfampin) from deer and bovine Lf are caused by differences in the sequence of amino acid.

L29: In order to uniformize the writing criteria throughout the manuscript, it is suggested that "78056 Da" be replaced by “78.1 kDa”.

L36-45: The second paragraph of the introduction is the framing paragraph of the article. It is suggested to put this one as the first paragraph and to detail a little bit more on the subject.

L25-45: In the introduction, I suggest to add a paragraph about the antibacterial values of these peptides in bovine.

L51-54: In the last paragraph of the introduction, the authors present the study hypothesis. I agree that, for the scientific method, it is extremely important to have a work hypothesis. However, it is more common (and “reader friendly”) to present the objectives of the study.

L235: In the table 1, is suggested the use of bold character in the red letters, in order to be more perceptible in the gray scale printing.

L329: Tale 2 - Some suggestions: (1) to replace “*” by “> 960 mg/mL”, the highest concentration of peptide used in the assay; (2) Why have the authors omitted the results of MBC analysis? (3) to improve the article, is it possible to show more information about quality control of lab/replicate data? Standard deviation? ANOVA output?

L334-353: Is it possible to improve the quality of the figures, or improve the contrast, in order to be more perceptible, the analysis performed by the authors?

Author Response

Manuscript: foods 1216765

1. This manuscript entitled “Amino acid sequences of Lactoferrin from red deer (Cervus elaphus) milk and antibacterial activity of its derived peptides Lactoferricin and Lactoferrapin” aims to compare: i) The primary structure between deer and bovine lactoferrin (Lf); ii) The characteristics of modelled 3-D structures of Lactoferricin (Lfcin) and Lactoferrapin (Lfampin) peptides obtained from deer and bovine Lf and; iii) the antibacterial activities of Lfcin and Lfampin peptides obtained from deer and bovine Lf.

The originality and relevance of this study was well exposed and consisted in the fact that “No studies to date have been carried out investigating the amino acid sequence of deer Lf” and the antibacterial activity of it derived peptides Lfcin and Lfampin. The topic was well introduced (but can be improved), the procedure clearly described, and results were properly presented in tables and graphs and adequately discussed, using several updated bibliographic references.

The discussion and the conclusions were well described.

Based on these judgments, I recommend the manuscript for publication in Foods after minor revisions.

We thank the Reviewer for their positive comments about our manuscript.

2. L19-21: In the Abstract section, the conclusion can be stronger if the authors present the reason for concluding that the differences of antibacterial activity against E. coli 20 ATCC 25922 of the peptides (Lfcin and Lfampin) from deer and bovine Lf are caused by differences in the sequence of amino acid.

The abstract has been modified to incorporate information on differences in the amino acid sequences of deer and bovine Lfcin and Lfampin (Table 1) and differences in antimicrobial activity (Table 2).

3. L29: In order to uniformize the writing criteria throughout the manuscript, it is suggested that "78056 Da" be replaced by “78.1 kDa”.

Revised.

4. L36-45: The second paragraph of the introduction is the framing paragraph of the article. It is suggested to put this one as the first paragraph and to detail a little bit more on the subject.

Done.

5. L25-45: In the introduction, I suggest to add a paragraph about the antibacterial values of these peptides in bovine.

Additional information has been added.

6. L51-54: In the last paragraph of the introduction, the authors present the study hypothesis. I agree that, for the scientific method, it is extremely important to have a work hypothesis. However, it is more common (and “reader friendly”) to present the objectives of the study.

The text has been modified.

7. L235: In the table 1, is suggested the use of bold character in the red letters, in order to be more perceptible in the gray scale printing.

Done.

8. L329: Tale 2 - Some suggestions: (1) to replace “*” by “> 960 mg/mL”, the highest concentration of peptide used in the assay;

This has been changed as requested.

(2) Why have the authors omitted the results of MBC analysis?

The MBC values were found to be the same as the MIC values. The following text has been added to the title of Table 2:

“The MBC values were found to be the same as that of the MIC values.”

(3) to improve the article, is it possible to show more information about quality control of lab/replicate data? Standard deviation? ANOVA output?

All assays were performed in triplicate as mentioned in the text at the end of section 2.6 in the original manuscript. The statistical analysis of the data in Table 2 has been added.

9. L334-353: Is it possible to improve the quality of the figures, or improve the contrast, in order to be more perceptible, the analysis performed by the authors?

Done.

The manuscript has been proof read by fluent English speakers.